# A modified weighted log-rank test for confirmatory trials with a high proportion of treatment switching

**José L. Jiménez** **[1]\*, Julia Niewczas[2], Alexander Bore[2], Carl-Fredrik Burman[2]**

**1** Global Drug Development, Novartis Pharma A.G., Basel, Switzerland, **2** Statistical Innovation, Data Science & AI, AstraZeneca R&D, Gothenburg, Sweden

\* jose_luis.jimenez@novartis.com

**Data Availability Statement:** The data used in this study, including the R code, is available within the manuscript and its Supporting information files.

## Abstract

In confirmatory cancer clinical trials, overall survival (OS) is normally a primary endpoint in the intention-to-treat (ITT) analysis under regulatory standards. After the tumor progresses, it is common that patients allocated to the control group switch to the experimental treatment, or another drug in the same class. Such treatment switching may dilute the relative efficacy of the new drug compared to the control group, leading to lower statistical power. It would be possible to decrease the estimation bias by shortening the follow-up period but this may lead to a loss of information and power. Instead we propose a modified weighted log-rank test (*mWLR*) that aims at balancing these factors by down-weighting events occurring when many patients have switched treatment. As the weighting should be pre-specified and the impact of treatment switching is unknown, we predict the hazard ratio function and use it to compute the weights of the *mWLR*. The method may incorporate information from previous trials regarding the potential hazard ratio function over time. We are motivated by the RECORD-1 trial of everolimus against placebo in patients with metastatic renal-cell carcinoma where almost 80% of the patients in the placebo group received everolimus after disease progression. Extensive simulations show that the new test gives considerably higher efficiency than the standard log-rank test in realistic scenarios.

## 1 Introduction

Research in oncology has been increasing over the past years. This can be seen for example by an increased proportion of cancer trials registered at `clinicaltrials.gov`. In years 2007–2010, trials in oncology comprised 21.6% [1] while in 2017 the number rose to almost 35% of all registered trials [2]. In December 2018 the Food and Drug Administration (FDA) in the US updated their guidance "Clinical Trial Endpoints for the Approval of Cancer Drugs and Biologics" [3] with recommendations on the choice of appropriate endpoints when performing clinical trials in oncology. Overall survival (OS) is considered by regulatory authorities as the most relevant and reliable clinical endpoint. However, it usually requires a long follow-up which could delay approval of a beneficial treatment. It is therefore common to use

Additionally, the R code can be accessed at https://github.com/borealexander/tslrt.

**Funding:** JJ is employed by Novartis Pharma A.G. and JN, AB, CF are employed by AstraZeneca. The funders provided support in the form of salaries for authors, but did not have any additional role in the preparation of the manuscript. The specific roles of these authors are articulated in the 'author contributions' section. Moreover, the views expressed in this publication are those of the authors and should not be attributed to any of the funding institutions, or organizations with which the authors are affiliated.

**Competing interests:** JJ is employed by Novartis Pharma A.G. and JN, AB, CF are employed by AstraZeneca. There are no patents, products in development or marketed products associated with this research to declare. This does not alter our adherence to PLOS ONE policies on sharing data and materials.

a surrogate endpoint that is a good predictor of OS. One endpoint that is frequently used in trials and allows for accelerated approval is progression-free survival (PFS). In some situations, PFS might be enough to obtain traditional regulatory approval [4–7]. Carneiro et al. [8] published an overview of accelerated and traditional regulatory approvals in oncology. However, in most cases, the traditional approval can only be obtained after showing efficacy also on OS and the accelerated approval might be withdrawn if the due diligence is not demonstrated [3].

For ethical reasons, a patient may switch treatment after disease progression. Such switching will not have an impact on PFS, but it may have a high impact on OS. Patient crossover might result in a diluted effect on OS, decreasing the power of the study. See for example the RECORD-1 trial [9] presented in Section 2. Even in the presence of patient crossover, it is preferred by the regulatory authorities to perform an intention-to-treat (ITT) analysis [10] where treatment groups are compared as originally randomized. Using ITT as the primary analysis has been criticized as it tends to underestimate the true treatment effect [10]. On the other hand, ITT analysis is robust in the sense that it is unlikely that any bias would inflate the type-I error rate.

It may be a clinically relevant question to estimate the efficacy that would have been observed if no patients had switched in the study. Alternative approaches have been proposed in the literature to address the issues of estimating the hazard ratio in the presence of treatment switching. These methods focus on the issues of estimation and bias mainly in the context of Health Technology Assessments (HTAs). They include the use of a per protocol analysis that either censors patients at the time of switching, or removes them from the analysis set [10–12], which can result in a selection bias. More complex methods include inverse probability of censoring weighting (IPCW) [13], rank-preserving structural failure time (RPSFT) model [14] and two-stage adjustments [15, 16] that were further simplified [17, 18]. Advantages and disadvantages of all methods have been discussed by researchers and regulatory authorities [10–12, 19, 20]. These methods have proven to have a smaller bias than simply using the ITT method under some circumstances [17]. Latimer et al. [18] showed that RPSFT model, IPCW and two-stage adjustment are likely to provide good approximations of the true treatment effect as long as the proportion of patients switching treatments is moderate. However, EMA [12] points out that "RPSFT models will typically not change the p-value, and while IPCW and 'two-stage' methods might, confidence intervals for all three methods tend to be wide", meaning that while estimates of hazard ratio might be less biased, the power of the trial will not be increased. Furthermore, it is stressed that underlying assumptions of these methods cannot be proven to be true [12]. As noted above, it is still generally required to base the primary analysis on the ITT set of patients and use the alternative methods as complements to ITT [12, 17, 18]. EMA [12] stresses that "due to the uncertainties involved in the methods (. . .), such estimations should, at present, be used primarily as supportive or sensitivity analyses".

Lately, non-proportional hazards have been receiving a lot of attention since immuno-oncology agents present what is known as a delayed treatment effect, which violates the proportional hazard assumption [21]. In this case, a common model assumes lack of treatment differences at the beginning of the trial (i.e., the hazard ratio is equal to 1) and treatment differences after some unknown time point (i.e., the hazard ratio is no longer equal to 1). Alternatively, and perhaps more realistically, one may assume that the relative treatment efficacy is gradually increasing over time. Methods addressing the problem of power loss in that context include the restricted mean survival time (RMST) [22], landmark analysis [23], accelerated failure time model [24], weighted Kaplan-Meier statistics [25], weighted log-rank tests (e.g. with Fleming and Harrington class of weights [26]), Max Combo test (taking the maximum value of a set of different weighted log rank tests) [27] and the "modestly weighted log-rank test" [28]. Another approach could be to simply increase the sample size in the trial accounting

for the delayed effect, but this will be inefficient and in many situations infeasible, considerably increasing the cost and length of the study.

It is well know that under proportional hazards, the log-rank test (*LR*) is optimal among all tests based on the order of events (and censoring) [29, 30]. In the presence of delayed efficacy, though, the weighted log-rank test shows superiority over the *LR* test in situations where the experimental arm is in fact better than the control. The test assigns a small weight at an early time in the study, where no differences are expected, and a larger weight for later time points, where survival curves are expected to separate. However, it was shown by Magirr and Burman [28] that the weighted log-rank test (*WLR*) does not control the type-I error rate under some scenarios when the experimental arm performs worse than the control. Treatment switching induces non-proportional hazards, where the hazard ratio increases towards the end of the trial and dilutes power. Hence, a *WLR* test with decreasing weights can then be used to increase power [31].

The focus of this article is on non-proportional hazards induced by treatment switching. However, we encourage interested readers to see [32–37] for interesting discussions and methods tailored for other sources of non-proportional hazards (i.e., crossing hazards or delayed effects).

In this manuscript we propose a modified weighted log-rank (*mWLR*) test where the downweighting depends on how much treatment switching is expected in a setting where patients from the control arm are allowed to switch treatment after disease progression. The article is divided into the following sections. In Section 2 we introduce a motivating example based on the RECORD-1 trial. Section 3 presents the proposed *mWLR* test. In Section 4 we present the simulation set-up we use to test the proposed *mWLR* test. In Section 5, we provide the results of the simulations where the *mWLR* test is compared with *LR* and other alternative tests. In Section 6 we discuss the main conclusions, the weaknesses and strengths of our proposal as well as further research.

## 2 Motivating example: The RECORD-1 trial

In this section we introduce a case study. It will be used as a realistic scenario in which we can test the performance of our proposal and compare it with other methods. It is, however, not within the scope of this article to re-analyze or make new clinical interpretations of the data. RECORD-1 was a phase III trial that examined the impact of everolimus (Afinitor; Novartis Pharmaceuticals Corporation, East Hanover, NJ) on the primary endpoint of PFS, and the secondary endpoints of OS and safety in metastatic renal-cell carcinoma (mRCC) patients, after treatment failure on sunitinib or sorafenib. It was a double-blind, multicenter study with patients randomized to receive either everolimus (n = 277) or placebo (n = 139) in a 2:1 ratio. Further details of the study design as well as main results have been presented in Escudier et al. [9] and Korhonen et al. [38].

One important aspect of the trial is that placebo patients had the opportunity to receive everolimus after disease progression, since existing literature already supported the antitumor activity of everolimus and another mTOR inhibitor temsirolimus in this indication [39, 40]. The study design therefore allowed for crossover to open-label everolimus following progression for patients randomized to placebo. In fact, 106/139 placebo patients did switch to open-label everolimus after disease progression. Furthermore, when the study was unblinded on February 28, 2008, a planned interim analysis showed significant superiority of everolimus over placebo on the primary endpoint PFS (hazard ratio (HR) 0.33; 95% confidence interval (CI) 0.25–0.43; *LR* test p-value < 0.001). After this time, five of the remaining six patients still receiving placebo switched to open-label everolimus, yielding a total of 111/139 placebo

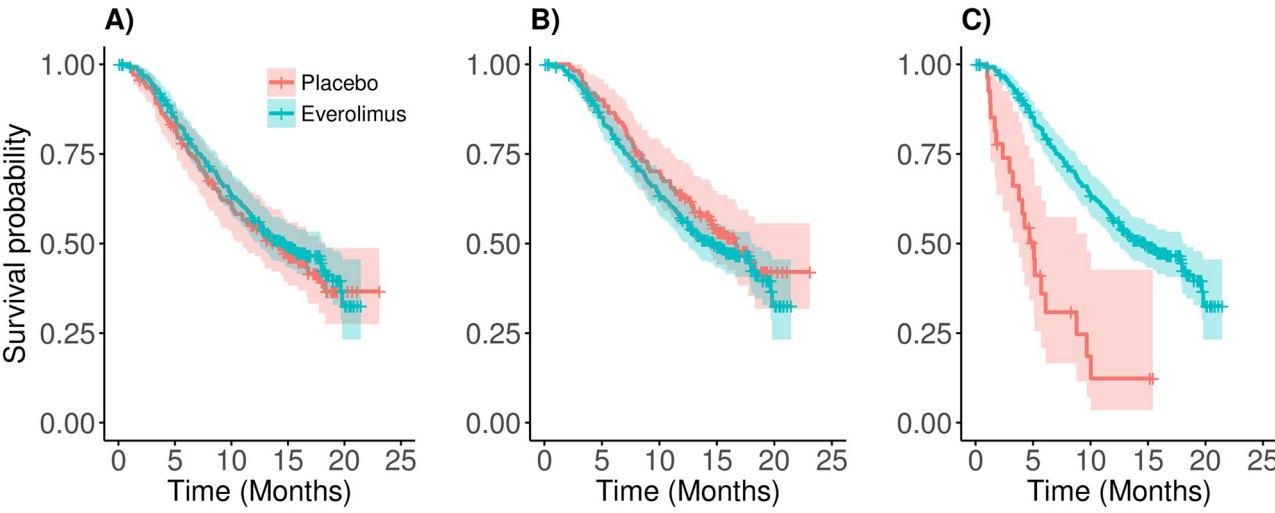

**Fig 1. Kaplan-Meier curves from the ITT analysis including all patients (image A), with only treatment switchers in the placebo arm (image B), and with only non-switchers in the placebo arm (image C).** Sample size in the placebo arm is equal to 139, 111 and 28 patients in plots A, B and C, respectively. Sample size in the Everolimus arm is equal to 277 patients in all plots.

patients that switched to everolimus. Patients were further followed up for survival until November 15, 2008. The ITT analysis of OS at this cut-off date yielded a HR of 0.87, which was in favor of everolimus, although it was not statistically significant (95% CI 0.65–1.15; one-sided p-value = 0.162). What makes this case study interesting in our context is that the OS results may have been biased due to the large extent of treatment switching.

In Fig 1A, we present the OS curves from the ITT analysis, in Fig 1B the OS curves with only switchers in the placebo arm, and in Fig 1C the OS curves with only non-switchers in the placebo arm. Without assumptions, the median OS for the placebo group cannot be directly estimated from the plots although they provide an insight of what impact the treatment switching could have had on OS in this trial. The median was in fact estimated using the rank-preserving structural failure time (RPSFT) model [38] and the crossover-adjusted median OS estimate was then close to 10 months.

## 3 Methods

### 3.1 The log-rank (*LR*) test

Let $S(t)$ be the probability of survival at time $t \geq 0$ and be defined by $S(t) = 1 - F(t)$, where $F(t)$ is a differentiable cumulative distribution function. Let $f(t)$ be the corresponding probability density function. The hazard function can then be defined as $h(t) = f(t)/S(t) = -S'(t)/S(t)$.

Assume now that we have a clinical trial with a control arm and an experimental arm. The corresponding survival and hazard functions are $S_0(t)$ and $S_1(t)$, and $h_0(t)$ and $h_1(t)$, respectively. We test the following (one-sided) hypothesis:

$$H_0 : S_0(t) = S_1(t) \ \forall \ t \quad \text{vs.} \quad H_1 : S_0(t) < S_1(t) \ \exists \ t, \tag{1}$$

to see if we have an effect on the experimental treatment arm. In clinical trials it is common practice to use the hazard ratio (i.e., the ratio between the hazard functions of each treatment) to quantify treatment differences. The hazard ratio function is defined as $\eta(t) = h_1(t)/h_0(t)$.

To test this hypothesis we may use the *LR* test. Let $t_1 < \cdots < t_k$ be the $k$ distinct, ordered event times. The number of patients at risk at time $t_j$ is denoted by $n_{i,j}$ with $n_j := n_{0,j} + n_{1,j}$. Let

$d_{i,j}$ denote the number of events on arm $i$ at time $t_j$ with $d_j := d_{0,j} + d_{1,j}$. The $LR$ test statistic is then defined as

$$U^{LR} = \sum_{j=1}^{k} \left( d_{0,j} - d_j \frac{n_{0,j}}{n_j} \right), \tag{2}$$

where the expression inside the sum describes the difference in actual and under $H_0$ expected number of events on the control arm at each distinct time. Under the null hypothesis, we would have $E[U^{LR}] = 0$. The variance of $U^{LR}$ is given by Brown [41] as

$$V(U^{LR}) = \sum_{j=1}^{k} \left( \frac{n_{0,j} n_{1,j} d_j (n_j - d_j)}{n_j^2 (n_j - 1)} \right). \tag{3}$$

For large sample sizes the test statistic $Z^{LR} = U^{LR} / \sqrt{V(U^{LR})}$ is normally distributed with mean 0 and variance 1 under the null hypothesis, by the central limit theorem. For a model with proportional hazards, meaning $\eta(t) = c$, where $c > 0$ is any constant, this unweighted $LR$ test is optimal (see Schoenfeld [30]) and power will increase with sample size. However, this is not the case under the presence of treatment switching.

In Fig 2 we present a simple example that shows how the power behaves depending on the number of events both under the proportional hazards model and under the presence of treatment switching. More information about how the model is set up will be described in Section 3.3. Under proportional hazards we see that, using the $LR$ test, the power increases with the

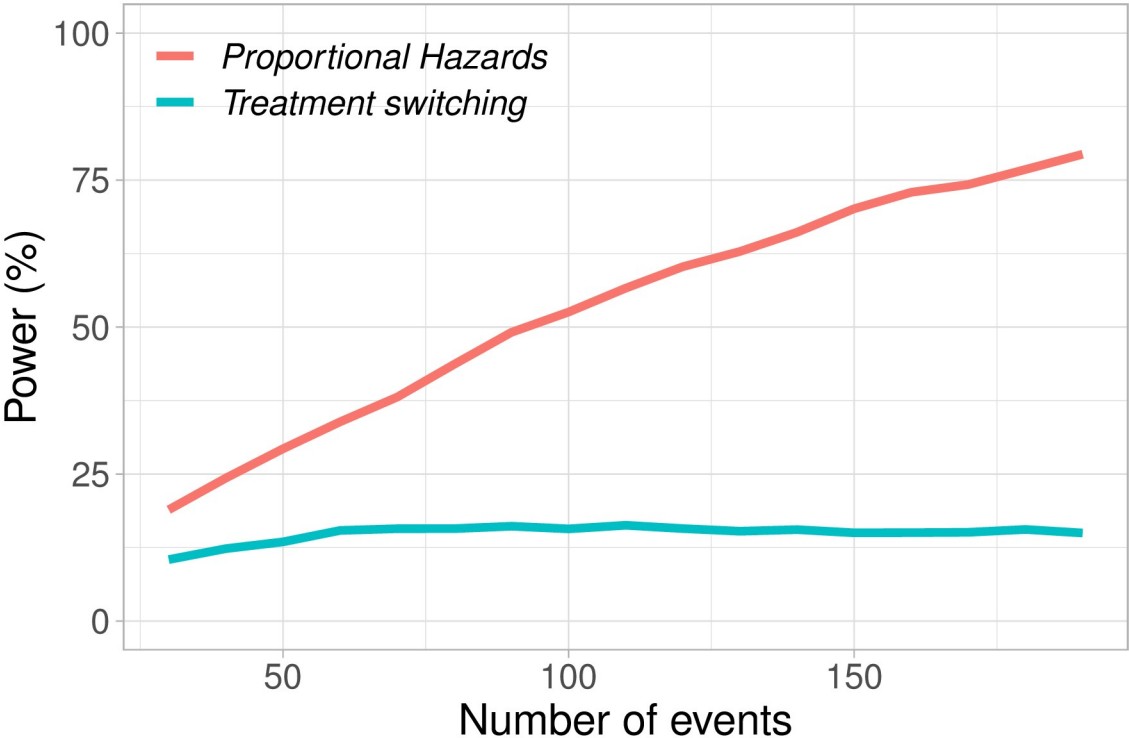

**Fig 2. Power with the log-rank ($LR$) for different total number of events under two different models: The proportional hazards model and the exponential progression switching model presented in Section 3.3 where patients switch treatment after disease progression with probability $p$.**

number of events. However under treatment switching, using the *LR* test, we see that the power increases up to a point, and then decreases.

## 3.2 Weighted log-rank (*WLR*) tests

Under the presence of treatment switching and following non-proportional hazards, the standard *LR* test is suboptimal. An alternative is the *WLR* test, defined as

$$U^{WLR} = \sum_{j=1}^{k} w_j \left( d_{0,j} - d_j \frac{n_{0,j}}{n_j} \right), \tag{4}$$

with variance

$$V(U^{WLR}) = \sum_{j=1}^{k} w_j^2 \left( \frac{n_{0,j} n_{1,j} d_j (n_j - d_j)}{n_j^2 (n_j - 1)} \right). \tag{5}$$

The test statistic is defined as $Z^{WLR} = U^{WLR} / \sqrt{V(U^{WLR})} \sim N(0,1)$ under the null hypothesis [26]. By setting the weights $w_j = 1$ (or any constant) we will get the standard (and unweighted) *LR* test.

Under the presence of treatment switching one expects a higher treatment effect in the beginning of the study that will decrease as the study progresses. Intuitively, by down-weighting late events, where treatment switching is expected to be high, we would achieve higher power than the standard *LR* test. For instance, the well known Fleming and Harrington class of weights [26] have been receiving a lot of attention over the last years, in particular with the development of immuno-therapy (see e.g., Jiménez et al. [42]), since they allow to down-weight early or late event using the estimated pooled survival function $\hat{S}(t)$.

## 3.3 A modified weighted log-rank (*mWLR*) test based on exponential progression switching

In this section, we develop a *mWLR* test that is tailored to a situation with considerable treatment switching. Under the assumption that we know the true hazard rate function, the optimal *LR* weights would be

$$w_j = -\log(\eta_j), \tag{6}$$

where $\eta_j$ represents the hazard ratio at time $t_j$.

In a regulatory setting the hypothesis test has to be pre-specified, therefore we propose to derive a hazard ratio model based on relevant clinical parameters and, through Eq (6), obtain a pre-specified weight function.

For simplicity, we assume exponential distributions for both progression and death. Let OS for patients receiving the control and experimental treatment be defined respectively as

$$S_0^{OS}(t) = \exp(-\lambda_0^{OS} \cdot t),$$
$$S_1^{OS}(t) = \exp(-\lambda_1^{OS} \cdot t). \tag{7}$$

A PFS event is defined as either a disease progression or a death. We use independent exponential distributions for these two components of the PFS event and define time to PFS as the minimum of time to progression or death. Our model does not depend on progression in the experimental group, as it is not affected by treatment switching, but we assume that time to

progression in the control group has survival functions

$$S_0^{\mathrm{P}}(t) = \exp(-\lambda_0^{\mathrm{P}} \cdot t). \tag{8}$$

As the competing progression and death risks are assumed to be independent and constant, the probability $r$ that a patient in the control group has a progression before dying is

$$r = \frac{\lambda_0^{\mathrm{P}}}{\lambda_0^{\mathrm{P}} + \lambda_0^{\mathrm{OS}}}. \tag{9}$$

The total PFS hazard is the sum of the component hazards, hence

$$\lambda_0^{\mathrm{PFS}} = \lambda_0^{\mathrm{P}} + \lambda_0^{\mathrm{OS}}. \tag{10}$$

For the RECORD-1 trial, as for most other oncology phase III trials, the medians $m_i^{\mathrm{PFS}}$ and $m_i^{\mathrm{OS}}$ for PFS and OS respectively, are provided in the main publication [9]. We have that

$$\lambda_0^{\mathrm{OS}} = \log(2)/m_0^{\mathrm{OS}}, \tag{11}$$

and

$$\lambda_0^{\mathrm{P}} = \lambda_0^{\mathrm{PFS}} - \lambda_0^{\mathrm{OS}} = \log(2)/m_0^{\mathrm{PFS}} - \log(2)/m_0^{\mathrm{OS}}. \tag{12}$$

Therefore if $\lambda_0^{\mathrm{P}} = \log(2)/m_0^{\mathrm{P}}$, it follows that

$$m_0^{\mathrm{P}} = \frac{m_0^{\mathrm{PFS}} \cdot m_0^{\mathrm{OS}}}{m_0^{\mathrm{OS}} - m_0^{\mathrm{PFS}}}. \tag{13}$$

Directly following a progression, patients in the control group are assumed to switch to experimental treatment with probability $p$. Thus, the total probability that a control arm patients will switch treatment before dying is defined as

$$
\begin{aligned}
q \quad &= P(\text{progression}) \times P(\text{patient switches treatment} \mid \text{progression}) = r \times p \\
&= \left(1 - \frac{m_0^{\mathrm{PFS}}}{m_0^{\mathrm{OS}}}\right) \times p.
\end{aligned} \tag{14}
$$

Note that this is the probability of the patient progressing and switching at some time point. Progressions occurring after censoring will not be observed in a trial. Thus, the proportion of patients switching treatment before a trial ends will often be somewhat less than $q$.

Given the characteristics of the RECORD-1 trial discussed in Section 2, a patient switching from control to experimental treatment is assumed to switch OS hazard from $\lambda_0^{\mathrm{OS}}$ to $\lambda_1^{\mathrm{OS}}$. This completes the model assumptions and we can now derive the hazard ratio function to obtain the proposed weights. The rationale of the model assumptions as well as alternative model choices are discussed in Section 6.

Patients randomized to the control group will belong, at a certain time $t$, to one of the following 4 categories: (i) non-progressed (np), (ii) progressed and having switched to experimental treatment (ps), (iii) progressed non-switched (pns), or (iv) deceased. The flow between the Markov states is shown in Fig 3 and the probabilities for the first 3 categories are given by the starting conditions $S^{np}(t) = 1$, $S^{ps}(t) = 0$ and $S^{pns}(t) = 0$, as well as by the following

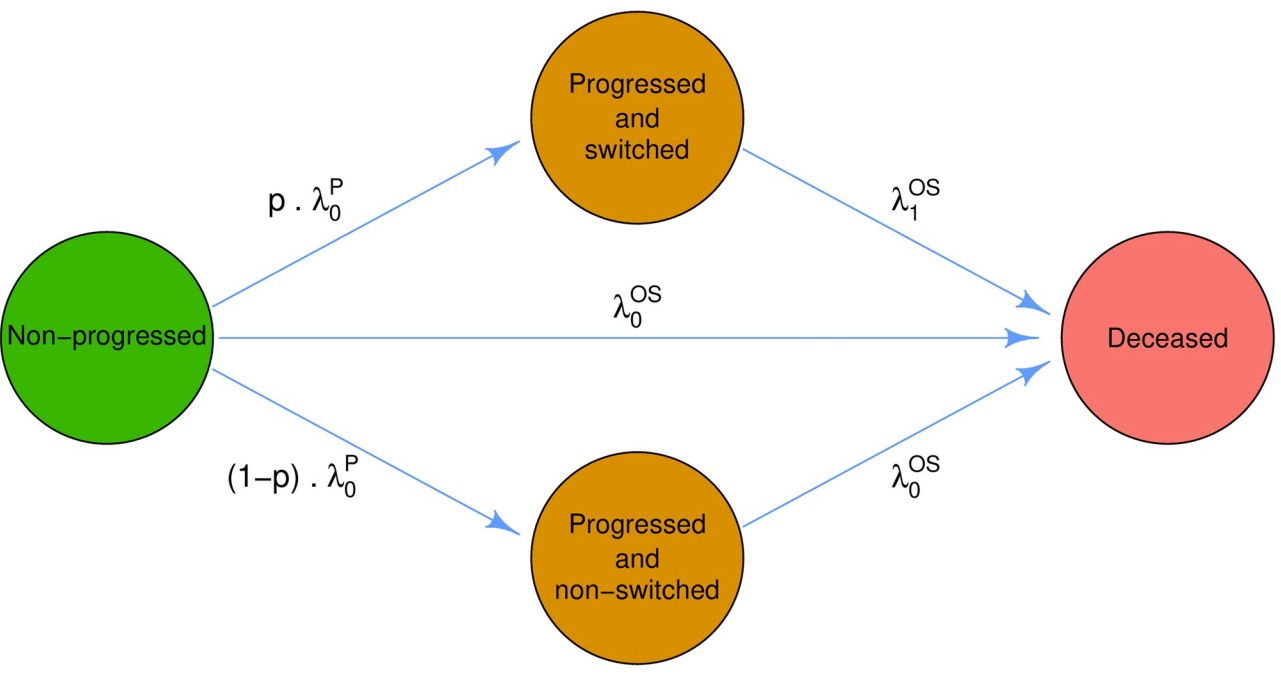

**Fig 3. Markov chain states and transition rates.**

differential equations:

$$\frac{dS^{np}(t)}{dt} = -(\lambda_0^{\mathrm{P}} + \lambda_0^{\mathrm{OS}}) \cdot S^{np}(t),$$

$$\frac{dS^{ps}(t)}{dt} = p \cdot \lambda_0^{\mathrm{P}} \cdot S^{np}(t) - \lambda_1^{\mathrm{OS}} \cdot S^{ps}(t), \tag{15}$$

$$\frac{dS^{pns}(t)}{dt} = (1-p) \cdot \lambda_0^{\mathrm{P}} \cdot S^{np}(t) - \lambda_0^{\mathrm{OS}} \cdot S^{pns}(t).$$

The solution to this system of differential equations is given by

$$S^{np}(t) = \exp(-(\lambda_0^{\mathrm{P}} + \lambda_0^{\mathrm{OS}}) \cdot t),$$

$$S^{ps}(t) = \frac{p \cdot \lambda_0^{\mathrm{P}}}{\lambda_0^{\mathrm{P}} + \lambda_0^{\mathrm{OS}} - \lambda_1^{\mathrm{OS}}} \cdot \left(\exp\left(-\lambda_1^{\mathrm{OS}} \cdot t\right) - \exp\left(-\left(\lambda_0^{\mathrm{P}} + \lambda_0^{\mathrm{OS}}\right) \cdot t\right)\right), \tag{16}$$

$$S^{pns}(t) = (1-p) \cdot \left(\exp(-\lambda_0^{\mathrm{OS}} \cdot t) - \exp(-(\lambda_0^{\mathrm{P}} + \lambda_0^{\mathrm{OS}}) \cdot t)\right),$$

and the total survival function for the control arm is therefore defined as

$$S_0(t) = S^{np}(t) + S^{ps}(t) + S^{pns}(t) =$$

$$= (1-p) \cdot \exp(-\lambda_0^{\mathrm{OS}} \cdot t) + p \cdot \frac{\lambda_0^{\mathrm{P}} \cdot \exp(-\lambda_1^{\mathrm{OS}} \cdot t) + (\lambda_0^{\mathrm{OS}} - \lambda_1^{\mathrm{OS}}) \cdot \exp(-(\lambda_0^{\mathrm{P}} + \lambda_0^{\mathrm{OS}}) \cdot t)}{\lambda_0^{\mathrm{P}} + \lambda_0^{\mathrm{OS}} - \lambda_1^{\mathrm{OS}}}. \tag{17}$$

An obvious question is how this flow between Markov states translates into survival functions. In Fig 4A we provide an example of the composition of patients over time for each

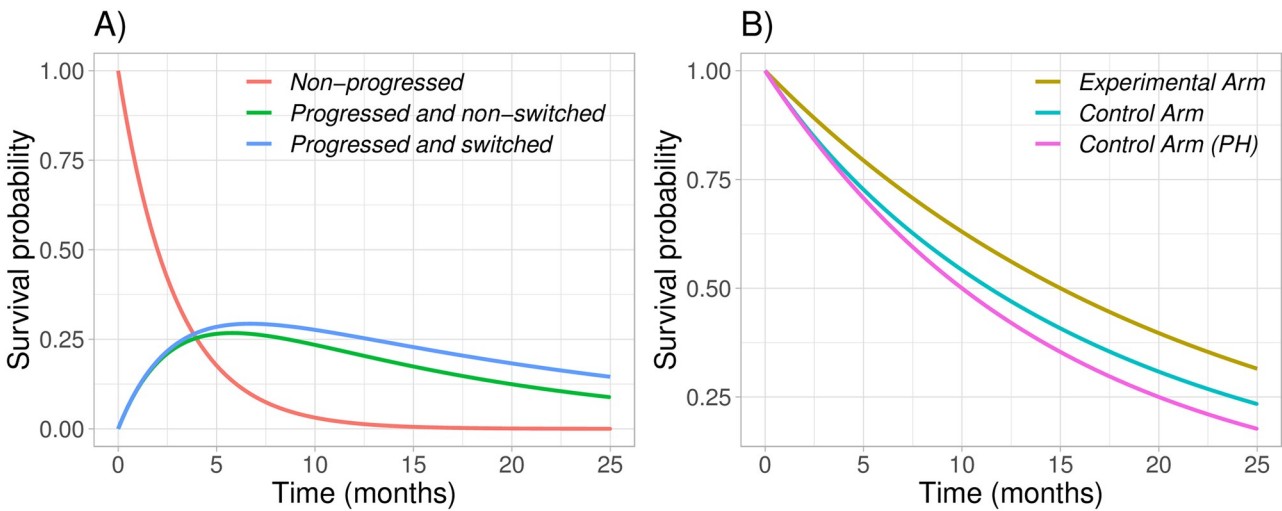

**Fig 4. Plot A shows the survival probability for non-progressed patients (red), progressed and non-switched patients (green) and progressed and switched patients (blue) assuming $p = 0.5$, $m_0^{\text{PFS}} = 2$, $m_0^{\text{OS}} = 10$, $m_1^{\text{OS}} = 15$.** In plot B, we show the resulting survival functions for the control (teal) and experimental (brown) arms, and also as a comparison, the control arm under proportional hazards (pink).

survival group as defined in Eq (16). In Fig 4B we show the resulting survival function for the control group (in teal). Moreover, in Fig 4B we also show the survival function for the experimental arm (in brown) as well as the survival function for the control group under proportional hazards (in pink) to make a visual comparison with the derived survival function for the control arm.

By definition, the hazard function for the control arm is $h_0(t) = -S_0'(t)/S_0(t)$. As the PFS rate is the sum of the progression and OS rates, $\lambda_0^{\text{PFS}} = \lambda_0^{\text{P}} + \lambda_0^{\text{OS}}$, it follows from Eq (17) that

$$h_0(t) = \frac{v_0(t) \cdot \lambda_0^{\text{OS}} + v_1(t) \cdot \lambda_1^{\text{OS}} + v_{0p}(t) \cdot \lambda_0^{\text{PFS}}}{v_0(t) + v_1(t) + v_{0p}(t)}, \tag{18}$$

where

$$
\begin{aligned}
v_0(t) &= (1-p) \cdot (\lambda_0^{\text{PFS}} - \lambda_1^{\text{OS}}) \cdot \exp(-\lambda_0^{\text{OS}} \cdot t) \\
v_1(t) &= p \cdot \lambda_0^{\text{P}} \cdot \exp(-\lambda_1^{\text{OS}} \cdot t) \\
v_{0p}(t) &= p \cdot (\lambda_0^{\text{OS}} - \lambda_1^{\text{OS}}) \cdot \exp(-\lambda_0^{\text{PFS}} \cdot t).
\end{aligned}
\tag{19}
$$

The hazard for the experimental arm simplifies to $h_1(t) = \lambda_1^{\text{OS}}$ and we can calculate the hazard ratio over time, $\eta(t) = \lambda_1^{\text{OS}}/h_0(t)$. Therefore, the weights defined in Eq (6) are computed as $w_j = -\log(\lambda_1^{\text{OS}}/h_0(t_j))$.

These weights will primarily depend on the assumed (conditional) treatment switching probability, $p$. Note that when $p = 0$ the model simplifies into a proportional hazards model that assigns constant weights. That is, the proposed *mWLR* test coincides with the standard unweighted *LR* test. A common time scale parameter will not alter the weights, in the sense that if all times and time parameters are multiplied by a constant $k$, all weights remain the same. The assumed relation between the median survival of experimental and control will have little impact on the test weights except for the obvious change in power and for the fact that the initial value of $w_j$ is proportional to the ratio between these medians of overall

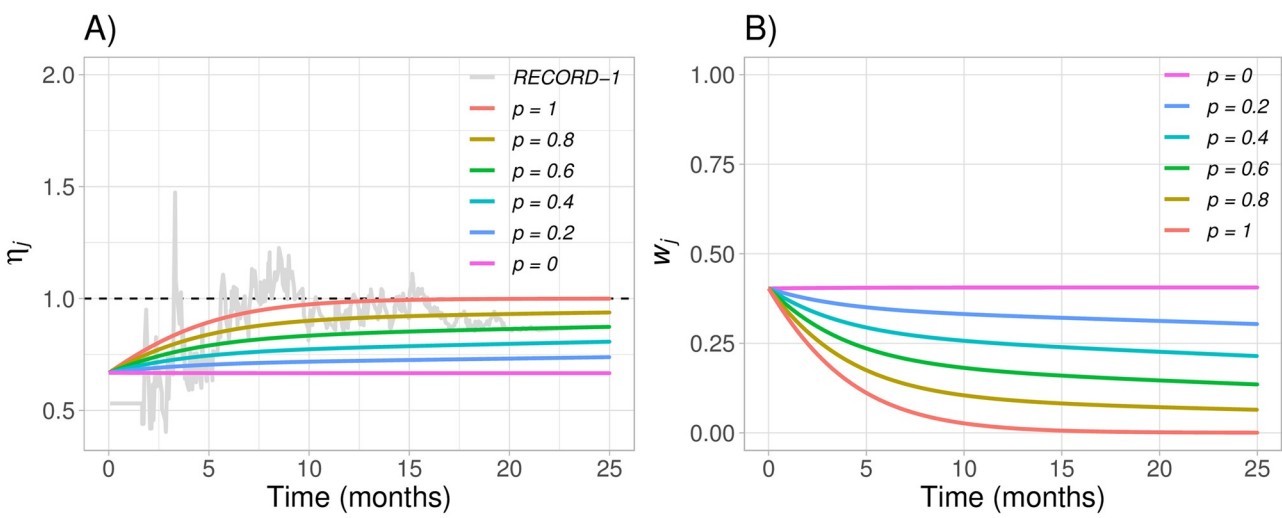

**Fig 5. Hazard ratio from the RECORD-1 trial and hazard ratio functions obtained with the model based on exponential progression switching (plot A), and corresponding weights functions from Eq (6) (plot B), for $p = (0, 0.2, 0.4, 0.6, 0.8, 1)$, $m_0^{\mathrm{OS}} = 10$ with $m_1^{\mathrm{OS}} = 15$ and $m_0^{\mathrm{PFS}} = 2$.** The weights using $p = 0$ are equivalent to those from the standard log-rank ($LR$) test.

survivals. The test will have some dependency on the relative medians for progression and overall survival, as the test differentiates between time points with different proportions of non-deceased patients that have switched treatment.

In Fig 5A, we present the hazard ratio functions produced by the proposed model assuming different values of $p$ together with the real RECORD-1 hazard ratio. In Fig 5B, we present the weight values obtained from Eq (6) for each of the hazard ratio functions presented in Fig 5A.

In this section we have proposed a model that uses clinically relevant parameters to build a realistic hazard ratio function that explains the impact of treatment switching in a clinical trial. In fact, in Fig 5A we see that when using $p = 1$ the proposed hazard ratio function closely approximates the hazard ratio function of the RECORD-1 trial. However, the true hazard ratio function is unknown when designing a trial, and the best we can do is build a "guess" based on prior clinical information from clinical trials with similar characteristics. In Sections 4 and 5 we implement an evaluation of the methodology presented in this section. To do so, we assume a true hazard ratio function and a pre-specified hazard ratio function that will be our "best guess". Let $p$ refer to the probability of treatment switching after disease progression from the true hazard ratio function and $p'$ refer to the probability of treatment switching after disease progression from the assumed (or "guessed") hazard ratio function.

## 4 Simulation study set-up

In this section we conduct a comprehensive simulation study to investigate the operating characteristics of the $mWLR$ test with the hazard ratio model based on exponential progressions introduced in Section 3.3. The entire simulation study is based on a single scenario set-up where sample size, median OS, median PFS and total number of deaths are the same from those observed in the RECORD-1 trial. These values are presented in Table 1.

Note that, in Table 1, the median OS in the control group ranges from 5 to 10 months as it is not possible to obtain its real value from the RECORD-1 trial data given the impact of treatment switching. The recruitment data was not taken from the RECORD-1 trial data and patients are assumed to be enrolled uniformly during 12 months.

**Table 1. Median OS, median PFS, sample size, and total number number of deaths based on the RECORD-1 trial.**

|  | Control Arm | Experimental Arm |
|---|---|---|
| Median OS (months) | 5–10 | 15 |
| Median PFS (months) | 2 | 4 |
| Sample Size | 139 | 277 |
| Total number of deaths | 221 | |

One may think of this set-up as if we would be designing a clinical trial similar to RECORD-1 trial, after having observed the RECORD-1 trial results. In other words, we are designing a clinical with solid and reliable historical information that allows us to pre-specify a sensible hazard ratio function. The performance of the $mWLR$ test is compared with the performance of the standard $LR$ test in terms of power and efficiency. The empirical power for both tests is calculated as

$$\text{Power}^{LR} = \frac{1}{M} \sum_{i=1}^{M} \mathbb{1}\left(Z_i^{LR} > \Phi^{-1}(1-\alpha)\right),$$

$$\text{Power}^{mWLR} = \frac{1}{M} \sum_{i=1}^{M} \mathbb{1}\left(Z_i^{mWLR} > \Phi^{-1}(1-\alpha)\right),$$

(20)

and the relative efficiency between $mWLR$ and $LR$ as

$$\text{Efficiency}_{LR}^{mWLR} = \left(\frac{\Phi^{-1}(1-\alpha) + \Phi^{-1}(\text{Power}^{mWLR})}{\Phi^{-1}(1-\alpha) + \Phi^{-1}(\text{Power}^{LR})}\right)^2,$$

(21)

where $\Phi^{-1}$ represents the quantile function of the standard Normal distribution, $\alpha = 0.025$ and $M = 10^4$ corresponds to the number of simulations implemented implemented in R [43] for each scenario. Eq (21) should be interpreted as the (empirical) efficiency of $mWLR$ with respect to $LR$ where values above 100% imply a better performance of the $mWLR$ test with respect to $LR$, and values below 100% imply a better performance of $LR$ with respect to the $mWLR$ test.

Let $\hat{\pi}$ denote the estimated power. As $M = 10^4$ simulation runs are performed for each point in power diagrams, the simulation error (95% confidence interval) is $\pm 1.96 \cdot \sqrt{\hat{\pi}(1-\hat{\pi})/M}$, at most $\pm 1.0$ percentage points.

One of the key characteristics of this methodology is that it relies on the pre-specification of a hazard ratio function that depends on prior values of median OS, median PFS and probability of switching. As presented in Fig 5, depending on $p$, the hazard ratio function has different shapes. However, if $p'$ is not close to $p$, the model may not work very well since the hazard ratio function would be misspecified. In the simulation study we primarily focus on evaluating the performance of the proposed model under the presence of a high proportion of treatment switching. However, we also make an evaluation in cases where $p$ and $p'$ are not close, and compare the results with those from the standard $LR$ test.

Moreover, in order to provide an entire overview of the model performance we also compare the $mWLR$ test, in a scenario with a high values of $p$, with the test based on the restricted mean survival time and the Max Combo test, which are known to have higher power than the standard $LR$ test under non-proportional hazards.

Note that we do not test on the real RECORD-1 data the proposed weight function built with the model presented in Section 3.3 since it is out of the scope of this article re-analyzing or making new clinical interpretations of the RECORD-1 trial data.

## 5 Results

### 5.1 Performance of the new test

In this section we present the results from the simulation set-up described in Section 4 that provides a scenario that can be considered similar to the RECORD-1 trial and hence realistic.

We start this performance evaluation by considering the extreme case where all patients in the control group switch treatment after disease progression ($p = 1$). If we take $p' = 1$, this is the scenario where $mWLR$ has the greatest benefit compared to the standard $LR$ test. Moreover, this scenario would not be so far from what was observed in the RECORD-1 trial, where 111 out of the 139 patients randomized to the control arm switched treatment, giving an estimate for $q$ of 80%. We don't know how many patients, in the control arm, died before progression but with reasonable assumptions regarding the competing risks for progression and death (see Eq (14)), $p$ can be expected to be considerably higher than $q$, although not exactly 1. Later on, we assess the test performance for $p' = p$ ranging from 0 to 1.

In Section 5.2 we assess the robustness of the proposed test when the degree of treatment switching is misspecified, with the test parameter $p'$ being different from $p$. Although the $LR$ test is currently the dominating analysis method, we compare $mWLR$ also against other alternatives, the Max Combo test and Restricted Mean Survival, in Section 5.3.

Throughout this section, we assume that median TTP is $m_0^{\mathrm{p}} = 2$ months in the control group. For the median OS, we assume $m_1^{\mathrm{OS}} = 15$ months for the experimental treatment, while we consider a range of values for patients on control treatment. With no treatment switching ($p = 0$) the *unaffected* hazard ratio for OS in our model would be $\mathrm{HR}_u = m_0^{\mathrm{OS}}/m_1^{\mathrm{OS}}$. Taking median survival on control ranging from $m_0^{\mathrm{OS}} = 5$ months to $m_0^{\mathrm{OS}} = 10$ months, the unaffected HR ranges from $HR_u = 5/15 \approx 0.33$ to $HR_u = 10/15 \approx 0.67$.

Fig 6 shows that $mWLR$ is much more powerful and therefore efficient than the $LR$ test when $p = p' = 1$ for different values of $m_0^{\mathrm{OS}}$. Obviously, the absolute increase in power depends on how large the power is for the $LR$ test. For example, when $m_0^{\mathrm{OS}} = 5$ ($HR_u \approx 0.33$), $LR$ power is equal to 96%, leaving limited room for further power increase. However, $mWLR$ reaches a power of 99%. When $HR_u = 0.5$, the absolute increase in power of $mWLR$ with respect to $LR$ is larger, going from 45% up to 66%. The efficiency with respect to $LR$ goes from 141% to 183% when going from $HR_u \approx 0.33$ to $HR_u \approx 0.67$. That is, the trial would have required a much lower sample size if the proposed $mWLR$ test would have been used instead of the standard $LR$ test.

In practice, $HR_u$ is unknown when planning the trial, so it is reasonable to consider the performance of the tests over a range of $HR_u$ values. If results on PFS are very convincing and treatment switching is high, it is not certain that regulators would require a formal statistical significance for OS. It is therefore of importance that the increase in efficiency with $mWLR$ would also lead to lower p-values even if neither test reaches statistical significance. For example, when $m_0^{\mathrm{OS}} = 10$ and power is relatively low, more than 99% of simulations gave a lower p-value for the $mWLR$ test than for standard $LR$.

The previous example with $p = 1$ is the most challenging scenario, but it is where $mWLR$ most clearly dominates the $LR$ test. With the correct treatment switching assumption (i.e., $p' = p$), $mWLR$ outperforms the standard $LR$ test for all values of $p$. However, as presented in Fig 7, the benefit is practically neglectable if the degree of treatment switching is low. In fact, we do not think that the $mWLR$ test is worthwhile if $p$ is known to be small, say $p \leq 0.4$. On the other

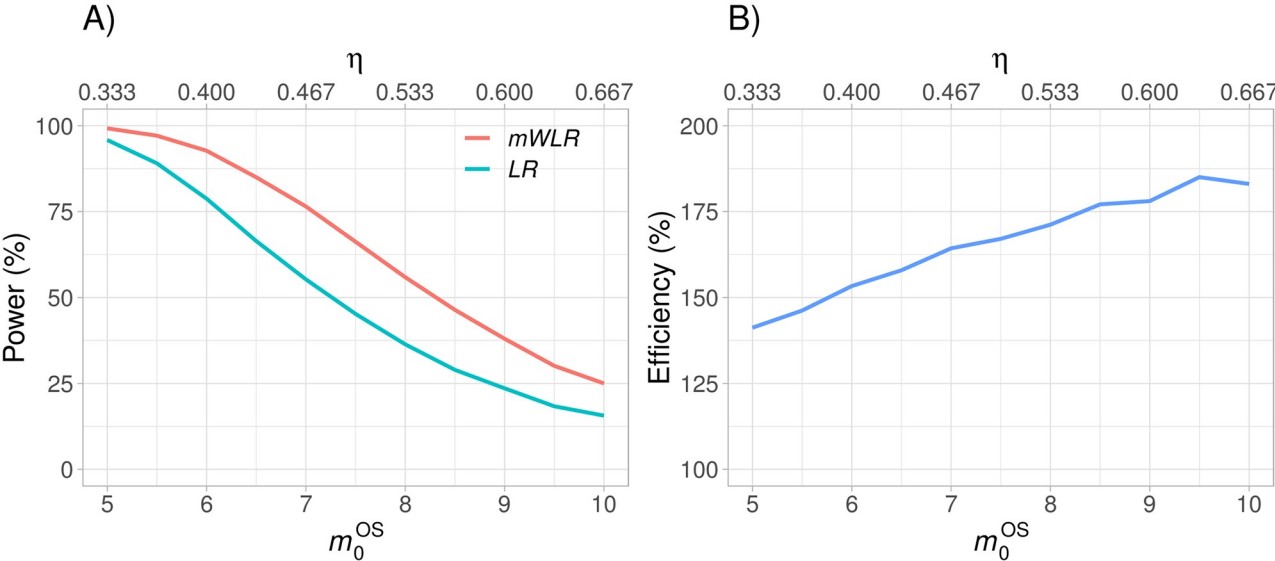

**Fig 6. Power of the modified weighted log-rank (*mWLR*) test and the log-rank (*LR*) test (plot A) and efficiency of the *mWLR* test with respect to the *LR* test (plot B) assuming $p = 1$ and $p' = 1$ for values of $m_0^{\mathrm{OS}}$ that range from 5 to 10 months and the corresponding hazard ratio $\eta$.**

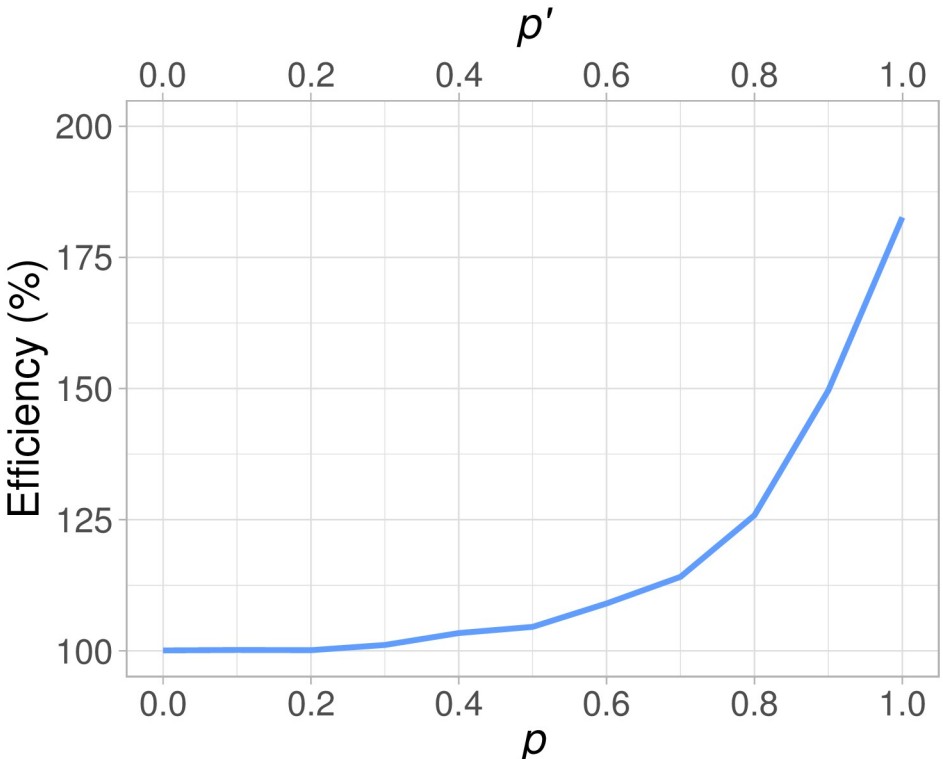

**Fig 7. Efficiency of the modified weighted log-rank (*mWLR*) test with respect to the log-rank (*LR*) test assuming matching values of $p$ and $p'$ (i.e., $(p = 0, p' = 0)$, $(p = 0.1, p' = 0.1)$, ..., $(p = 1, p' = 1)$) for a fixed value of $m_0^{\mathrm{OS}} = 10$ months.**

hand, the efficiency gain of approximately 5% when $p = 0.5$ is not neglectable since a 5% decrease in sample size may translate into high cost savings. An even clearer indication for using $mWLR$ is when the investigator team fears that $p \geq 0.75$. For example, when a trial is designed, the "best guess" of the probability of treatment switching after disease progression may be $p' = 0.6$, where the efficiency of $mWLR$ with respect standard $LR$ is about 109% if $p' = p$. However, the prediction of $p'$ may be quite uncertain, ranging from rather low treatment switching, where $LR$ would do well, up to perhaps $p' = 0.75$ (with a potential efficiency of around 120%) or even $p' = 0.9$ (with an efficiency of about 149%) if $p' = p$. Such situations, where $p'$ is uncertain when pre-specifying the analysis, will be further explored in the next subsection.

### 5.2 Robustness

As indicated by Fig 7, efficiency is increasing rapidly as $p$ increases, with a value of 183% when $p = 1$ and $p' = 1$. The downside is that, assuming $p' = 1$, $mWLR$ is only better than $LR$ when $p > 0.7$ and has an efficiency lower than 100% when treatment switching after disease progression is $p \leq 0.7$, as presented in Fig 8A and 8B. We could argue that $mWLR$ with $p' = 1$ is relatively robust when we are convinced that treatment switching will be very high. However, choosing a somewhat lower design parameter, $p'$, will give a more robust test if $p$ is not known to be 1.

By construction of the test, it is not surprising that $mWLR$ is the best test for a certain value of $p$ if designed with the matching treatment switching parameter (i.e., $p' = p$) as presented in Fig 9B, where the dot in each curve represent the value of $p'$ that maximizes efficiency of $mWLR$ with respect to $LR$ for a given value of $p$.

For a practical situation in a scenario with the characteristics presented in Section 4, a large expected treatment switching, but also a relatively large uncertainty around $p'$, one solution could be to assume $p' = 0.7$ for the following reasons:

1. The test is optimal if $p' = p$.

2. It is more efficient than the standard $LR$ if $p \geq 0.7$ with efficiency values that go from 114% $p = 0.7$ to 157% at $p = 1$ (see Fig 9A).

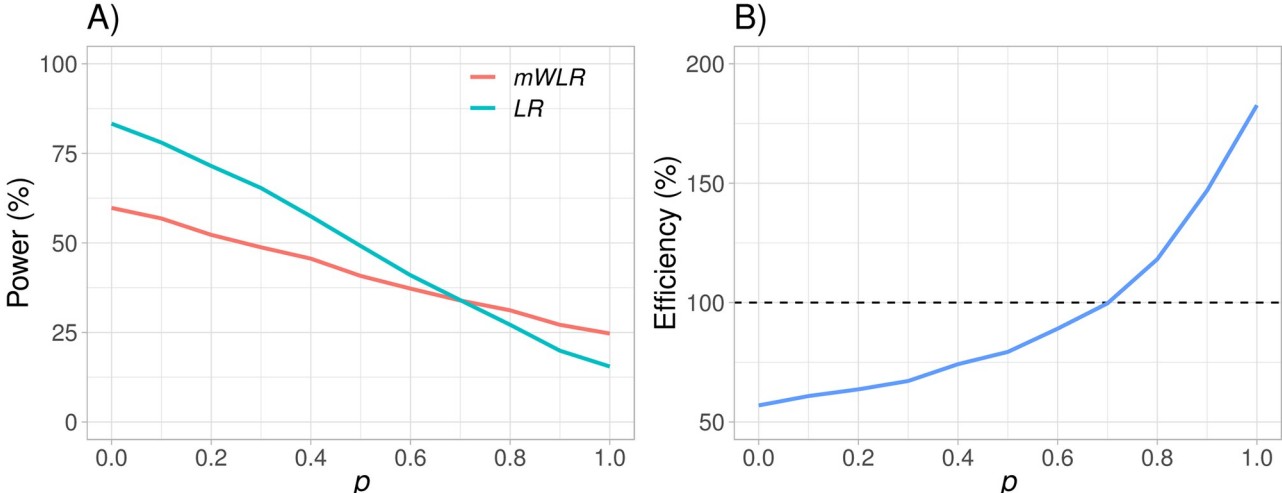

**Fig 8. Power of the modified weighted log-rank ($mWLR$) test and the log-rank ($LR$) test (plot A) and efficiency of the $mWLR$ test with respect to the $LR$ test (plot B) assuming a fixed value of $m_0^{OS} = 10$ months, a fixed value of $p' = 1$, and varying the value of $p$ between 0 and 1.**

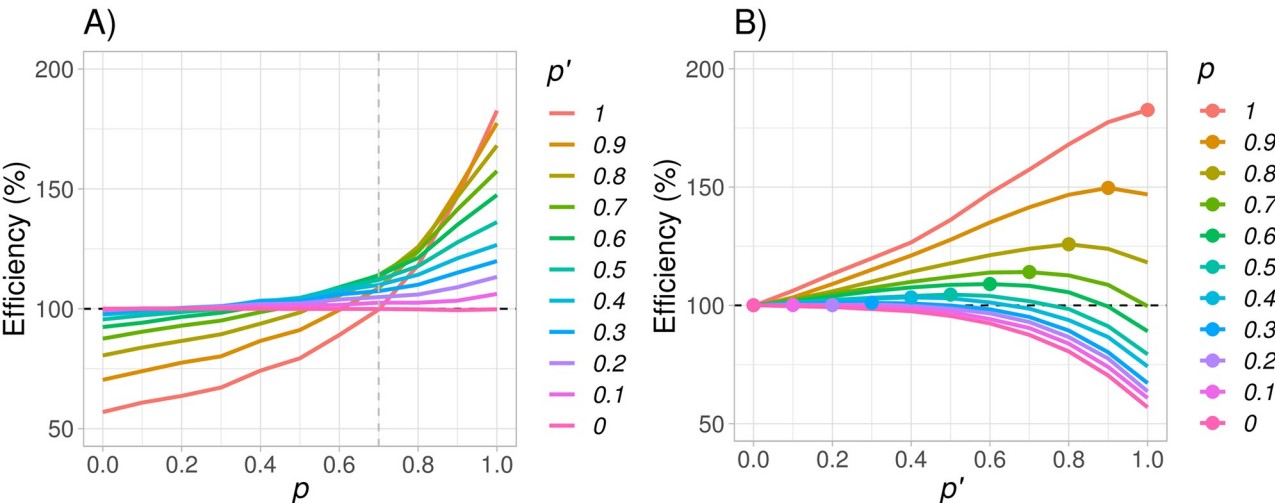

**Fig 9. Efficiency between the modified weighted log-rank (*mWLR*) test and the log-rank (*LR*) test assuming a fixed value of $m_0^{OS} = 10$ months and varying both *p* and *p′* between 0 and 1.** Values above 100% favor *mWLR* and values below 100% favor *LR*.

3. If $p < 0.7$, it would still be more efficient than standard *LR* for values of $p \geq 0.45$.

4. If $p < 0.45$, the loss would still be rather limited, especially for values of $p \geq 0.3$ where the efficiency is above 95%.

5. If $p = 0$, the test has an efficiency of 88%.

Thus, *mWLR* with $p′ = 0.7$ shows a balance between being robust to mid-degrees of treatment switching while having large efficiency for high-degree treatment switching with respect to the standard *LR* test.

As noted, the *relative* values of the medians are much more important for the results than the *absolute* values. The model is time-scale invariant. This means that, if all times, medians, etc., were multiplied with a factor 2, say, the power would be the same. Follow-up and inclusion time also has to be expanded for this to hold exactly, but as long as maturity (the fraction of patients followed to death) does not change much, the impact of these times is rather limited. Of greater interest is what happens if median time to progression is changed relative to median OS. Supplementary material contains replications of Fig 9B when $m_0^{PFS}$ is 1 and 4 months, respectively, instead of 2 as in the main model (see S1 and S2 Figs in S1 File respectively). When $m_0^{PFS} = 1$, patients progress faster with respect to $m_0^{PFS} = 2$, which translates into higher values of *q* (i.e., a larger proportion of patients that actually switch after disease progression before dying) and therefore a higher efficiency of *mWLR* with respect to *LR* compared to the one observed with $m_0^{PFS} = 2$. In contrast, when $m_0^{PFS} = 4$, patients progress slower which translates into lower values of *q* and therefore a lower efficiency of *mWLR* with respect to *LR* compared to the one observed with $m_0^{PFS} = 2$. However, these differences are in line with how the test is constructed and we can say the conclusions obtained using $m_0^{PFS}$ equal to 1 and 4 are qualitatively the same to the conclusions obtained with $m_0^{PFS} = 2$.

We also evaluate the robustness of the proposed *mWLR* test when the underlying time-to-event distribution is not exponential. More precisely, we use a Weibull distribution with shape parameter *k* ranging from 0.5 to 1.5 and scale parameter *λ* selected so the distribution yields a desired median OS or median PFS. With this distribution, it is possible to show that the hazard function of each treatment group decreases if $k < 1$, increases if $k > 1$ and is constant if $k = 1$.

In S3 Fig in S1 File we assess the power of $mWLR$ and $LR$ varying the value of $k$ in the same setting presented in Fig 6. Results show that $mWLR$ performs slightly worse than $LR$ for values of $k \leq 0.7$. In contrast, when $k > 0.7$, our proposal outperforms $LR$. This assessment shows that $mWLR$ is sensitive to increments/decrements of the hazard function of each arm and thus a careful evaluation of the expected hazard function of each arm is advised since the expected power of $mWLR$ may change.

### 5.3 Comparison with other methods

In this section we provide a comparison with two methods that have been receiving quite a lot of attention in the last years given their good performance under non-proportional hazard with respect to the standard $LR$ test: the Max Combo test [27, 44] and the test based on the restricted mean survival time [22].

It is important to mention that the test based on the restricted mean survival time is highly depending on the truncation time. Hence, in order to have an objective and fair comparison between these tests, the truncation time for the test based on the restricted mean survival time is linked to the data and is pre-specified as the minimum of the maximum observed event or censored time of each arm (i.e., minimax observed time).

With respect to the Max Combo test and following Roychoudhury et al. [45], we implement it considering the maximum of four correlated Fleming-Harrington class of weights: ($\rho = 0$, $\gamma = 0$), ($\rho = 1$, $\gamma = 0$), ($\rho = 0$, $\gamma = 1$) and ($\rho = 1$, $\gamma = 1$).

This comparison is made using the same set-up as in Section 5.1 which is realistic and similar to the RECORD-1 trial. The efficiency of $mWLR$ with respect to the test based on the restricted mean survival time (Fig 10A), shows that when the recommended value $p' = 0.7$ is used, in a scenario with these characteristics, $mWLR$ is more efficient that the test based on the restricted mean survival time for $p \geq 0.45$, reaching an efficiency of 137% when $p = 1$, and 112% when $p = 0.7$ where the test is optimal. Moreover, the efficiency loss for $p < 0.45$ is rather limited with an efficiency of 90% even when $p = 0$.

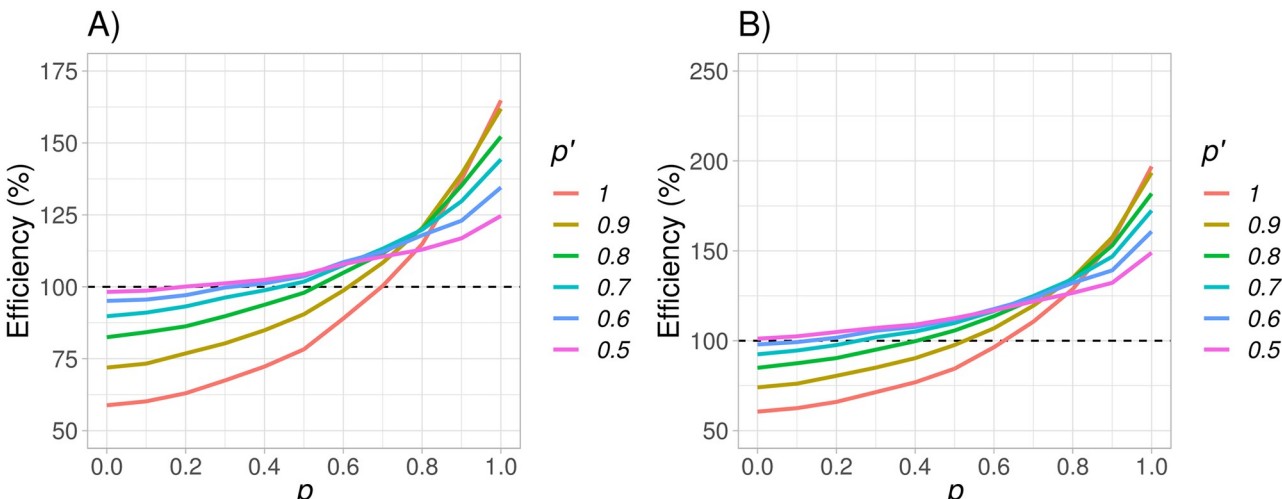

**Fig 10. Efficiency between the modified weighted log-rank test ($mWLR$) with respect to the test based on the restricted mean survival time (plot A), and efficiency between $mWLR$ test with respect to the Max Combo test (plot B) assuming a fixed value of $m_0^{OS} = 10$ months varying $p$ between 0 and 1, and $p'$ between 0.5 and 1.** Values of efficiency over 100% favor the $mWLR$ test and values below 100% favor the test based on the restricted mean survival time or the Max Combo test.

The efficiency of $mWLR$ with respect to the Max Combo test (Fig 10B), shows that in a scenario with these characteristics, using the recommended value $p' = 0.7$, $mWLR$ is as good as, or more efficient than the Max Combo test for all values of $p$, reaching an efficiency of almost 200% when $p = 1$. For values of $p \leq 0.3$ however, the performance between $mWLR$ and Max Combo is similar with efficient values below 105%.

Overall, these results are in line with the results obtained in Section 5.2 where the test is fairly robust for $p' = 0.7$ also in comparison with other testing alternatives particularly suitable for scenarios where the proportional hazards assumption does not hold.

## 6 Discussion

In this article we propose a new class of weighted log-rank ($WLR$) tests to be used when treatment efficacy is decreasing over time. The motivating application is when a substantial amount of patients in the control group switches during the clinical trial to a more effective treatment. This is, for ethical reasons, often occurring in phase III oncology clinical trials. However, the proposed test or another test based on the same idea may be applicable in other situations when there is a similar pattern of decreasing efficacy over time. One example outside of the treatment switching application is when the experimental treatment affects only one of several risk components. More precisely, if a trial is including patients with a recent stroke, a drug that is effective in preventing (new) strokes may show large benefit on all-cause mortality in an initial phase. However, the hazard ratio will gradually increase over time when the relative risk of stroke-related deaths starts to decrease and other causes of mortality start to be present in a large part of the total number of deaths.

According to the FDA [3], endpoints for later phase efficacy studies evaluate whether a drug provides a clinical benefit such as prolongation of survival or an improvement in symptoms. In oncology late phase clinical trials, overall survival (OS) is usually the preferred endpoint for final approval since it does not rely on any assumption, although progression-free-survival (PFS) is often accepted for conditional approval, awaiting direct evidence of a survival benefit.

The FDA [46] also acknowledges that in most trials, some patients may not receive the treatment assigned by randomization because of poor response, improvement or worsening of disease, or high toxicity among other reasons. In general, informative dropout may be of concern even if it occurs before the initiation of treatment as it can cause a distortion of the results. However, despite the potential treatment effect dilution that an intent-to-treat (ITT) analysis may cause, this type of analysis is the gold standard in confirmatory trials. The reason is that it ensures that the comparability of populations created by randomization is maintained and reduces the risk that bias will be introduced during the trial or during the analysis. Treatment switching obviously may distort what would have happened if patients would have been treated only with the drugs from the treatment arms in which they were randomized. However, even if it is possible to model what the relative efficacy would have been without treatment switching, such a model cannot be based solely on randomization since unknown factors will influence which patients from the control arm switch treatment.

In this article we are motivated by the RECORD-1 trial, a phase III clinical trial that compares placebo with everolimus in patients with metastatic renal-cell carcinoma where almost 80% of the patients switched from the placebo arm to receive everolimus after disease progression. Fig 1B shows that placebo patients who switched treatment had much longer average survival than those who did not switch treatment as presented in Fig 1C. However, this comparison is not randomization-based. One of the key questions is why some patients choose to switch, or not to switch, treatment? One could for example imagine that patients from the

placebo arm with particularly poor prognosis could receive palliative care instead of a new treatment with potential side effects. Techniques used to analyse observational data could be useful for example to determine the magnitude of an effect in patients actively taking a drug. However, a statistically significant ITT comparison between two randomized groups would provide more robust evidence of treatment efficacy, although sometimes the ITT analysis of OS is not feasible given ethical constraints. The methodology proposed in this article aims to increase the power in an ITT analysis under a high proportion of patients that switch treatment after disease progression.

The most common test used for confirmatory time-to-event clinical trials is the unweighted log-rank test (*LR*). However, given that nonparametric tests are relatively infrequent for primary analyses in other clinical trials, one may ask why *LR* is so popular for survival trials. One answer is that common parametric test alternatives often are relatively in-efficient [47]. If the hazard ratio is constant over time, the unweighted *LR* test is the most powerful test. Proportional hazards is a decent approximation in some cases, but there are many examples where this assumption does not hold. One area in which this assumption is clearly not met is immuno-therapy (see e.g., Rahman [48]). However, from our point of view, it is a mistake to think that the *LR* should have a general precedence because it is labelled as "unweighted". One may view the Wilcoxon test [49] as a weighted version of the *LR* test, but one could equally well view LR as a weighted version of Wilcoxon. The fact is that *LR* is equivalent to attributing a certain strictly decreasing "score" to each observation, depending on its rank order (see Leton and Zuloaga [50]). We argue that different scores (or *LR* weights) should be used when they can be pre-specified to give considerably higher power while strongly controlling the type-I error.

The Fleming-Harrington class of weights can be used with strictly decreasing weights under the presence of treatment switching. However, if weights are not strictly decreasing then type-I error is not controlled as showed by Magirr and Burman [28]. This also holds for the Max Combo test, which is an omnibus test with four different Fleming-Harrington test components. A difference between our proposal and the Fleming-Harrington class of weights is that our weights are functions of time, instead of functions of the estimated pooled survival function. A benefit is that it is more natural to model the effect under treatment switching in the time scale. Also, as seen in the simulation results presented in this article, our test mostly outperforms the Max Combo test.

Hypothesis tests should be complemented with clinically relevant estimates. The Kaplan-Meier curves, together with censoring patterns, give essentially all information, although more condensed measures are also valuable. Median survival and survival at, for instance, 2 years are clinically meaningful. Cure rate (if applicable) and otherwise (restricted) mean survival may have even greater bearing while also being clinically interpretable even under non-proportional hazards (see [35]). In contrast, *LR* tests, weighted or unweighted, can give estimates of an average hazard ratio or a parametric hazard ratio time function. This can be done by simply multiplying the number of patients at risk in the experimental arm with the tentative hazard ratio when calculating the *LR* statistic. The hazard ratio that makes the test statistic equal zero leads to the hazard ratio estimate. Strictly speaking, our method corresponds to a hazard ratio estimate. However, an estimated hazard ratio is difficult to interpret when hazards are meaningfully non-proportional.

In this article, we have promoted the use of pre-specified *LR* tests with non-increasing weights that correspond to the predicted hazard ratio function over time. We have developed one class of weights using prior information regarding median times to progression and to death, depending on actual treatment, as well as about the expected probability of treatment switching. For a concrete clinical trial, it may be possible to develop other models for the

hazard ratio function, better tailored to existing pre-clinical and early-phase clinical data, other indications, competitor data, clinical judgment etc. (see Burman and Wiklund [51]).

We also tested the robustness of our proposal under hazard ratio model misspecification. In other words, we have assumed an expected probability of treatment switching and tested its performance when the true proportion of patients after disease progression that switch does not match the expected probability of treatment switching. In this evaluation, we found that the performance is clearly dependent on the accuracy of the expected probability of treatment switching. When the degree of treatment switching, $p$ is uncertain, we recommend designing the modified test based on a slightly lower value of $p$ than the best guess estimate, to give higher robustness. The comparisons also evaluate a median OS misspecification in the control arm. In this case, the model is not very sensitive to the median OS value used to defined the hazard ratio function and there are not big differences in terms of performance. We have also evaluated the performance of $mWLR$ when the underlying time-to-event distribution is not exponential. For this analysis, we have employed a Weibull distribution and we have observed that our proposal is sensitive to increments/decrements of the hazard function of the treatment groups. Therefore, we advise to carefully evaluate the expected hazard function of the treatment groups.

Some possible extensions of the current work could consider its implementation in an adaptive setting, or situations where the probability of treatment switching depends on calendar time (when the experimental drug gets more available), OS hazards increasing after progression and/or depending on time of progression, and other distributions than exponential.

## Supporting information

**S1 File.**
(PDF)

**S1 Data.**
(ZIP)

## Author Contributions

**Conceptualization:** José L. Jiménez, Julia Niewczas, Alexander Bore, Carl-Fredrik Burman.

**Methodology:** José L. Jiménez, Julia Niewczas, Alexander Bore, Carl-Fredrik Burman.

**Software:** José L. Jiménez, Alexander Bore, Carl-Fredrik Burman.

**Supervision:** Carl-Fredrik Burman.

**Validation:** Julia Niewczas, Alexander Bore, Carl-Fredrik Burman.

**Visualization:** José L. Jiménez, Julia Niewczas, Alexander Bore, Carl-Fredrik Burman.

**Writing – original draft:** José L. Jiménez, Julia Niewczas, Alexander Bore, Carl-Fredrik Burman.

**Writing – review & editing:** José L. Jiménez, Julia Niewczas, Alexander Bore, Carl-Fredrik Burman.

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
