## [Decision Letter · Decision Letter 0]

11 Jun 2021

PONE-D-21-06687

A modified weighted log-rank test for confirmatory trials with a high proportion of treatment switching

PLOS ONE

Dear Dr. Jimenez,

Thank you for submitting your manuscript to PLOS ONE. After careful consideration, we feel that it has merit but does not fully meet PLOS ONE’s publication criteria as it currently stands. Therefore, we invite you to submit a revised version of the manuscript that addresses the points raised during the review process.

Each reviewer recommended minor revision.  All of the points should be addressable. 

We look forward to receiving your revised manuscript.

Kind regards,

Alan D Hutson

Academic Editor

PLOS ONE

Journal Requirements:

4.Thank you for stating the following in the Financial Disclosure section:

We note that one or more of the authors are employed by a commercial company: Biostatistical Sciences and Pharmacometrics and Statistical Innovation

Reviewers' comments:

Reviewer's Responses to Questions

**Comments to the Author**

1. Is the manuscript technically sound, and do the data support the conclusions?

Reviewer #1: Yes

Reviewer #2: Yes

2. Has the statistical analysis been performed appropriately and rigorously? 

Reviewer #1: Yes

Reviewer #2: Yes

3. Have the authors made all data underlying the findings in their manuscript fully available?

Reviewer #1: Yes

Reviewer #2: Yes

4. Is the manuscript presented in an intelligible fashion and written in standard English?

Reviewer #1: Yes

Reviewer #2: Yes

5. Review Comments to the Author

Reviewer #1: The authors highlighted the important topic of treatment switching in clinical trials which makes conclusions about treatment effects difficult and should therefore be addressed more often. I thank the authors for bringing up a new possibility to handle this iusse.

I have only minor comments:

- How about an adaptive design ? (to i.e. assume another underlying distribution or amount of treatment swichtching)

- What about the robustness against the assumed underlying distribution (i.e. exponential distribution might often not be given) did you consider Weibull, Gompertz, log-lormal distributed data?

- What about a situation of crossing hazards?

- Which effect measure is best to go along with your test? What (time-dependent? ) effect measure would you propose?

- Did you consider what estimand framework the in the your paper described setting/test would refer to?

- In formula (1) (test hypothesis): Where is S0(t)>S1(t) included? (Not clear whether you have a one or two-sided hypothesis)

Reviewer #2: It is well-written paper and topic is also very important. The authors proposed an alternative weight method for WLR and showed the advantage over existing methods by simulation study. A motivative example is re-analyzed and demonstrate the potential usefulness of the new proposed method. The idea is clearly explained.

The simulations are done only with the case matching to the motivation example. I am not sure if the method is still working better for other situations. Maybe additional simulation can help to identify when the mWLR has more advantage.

The efficiency measure in formula (21) is the squared ratio between two statistics. Is there a reason/literature to choose this efficiency (why squared)? Is the ration of two powers more natural?

Minor points:

The asymptotic property of WLR is fine with fixed weights while mWLR used the weights related to the outcome, the asymptotic property still holds true?

Figure 1, it is better to put sample sizes for plot A, B, C in figure’s caption.

6. PLOS authors have the option to publish the peer review history of their article (what does this mean?). If published, this will include your full peer review and any attached files.

Reviewer #1: No

Reviewer #2: No

---

## [Author Response · Author response to Decision Letter 0]

10 Sep 2021

Dear reviewers,

Thank you for the commentaries you have given to us. We have address them all and updated the manuscript accordingly. All our responses are located in the file "response_to_reviewers.pdf" and the changes are highlighted in blue in the file "manuscript_track_changes.pdf".

---

## [Decision Letter · Decision Letter 1]

15 Oct 2021

A modified weighted log-rank test for confirmatory trials with a high proportion of treatment switching

PONE-D-21-06687R1

Dear Dr. Jiménez,

We’re pleased to inform you that your manuscript has been judged scientifically suitable for publication and will be formally accepted for publication once it meets all outstanding technical requirements.

Kind regards,

Farzad Taghizadeh-Hesary

Academic Editor

PLOS ONE

Additional Editor Comments (optional):

Reviewers' comments:

Reviewer's Responses to Questions

**Comments to the Author**

1. If the authors have adequately addressed your comments raised in a previous round of review and you feel that this manuscript is now acceptable for publication, you may indicate that here to bypass the “Comments to the Author” section, enter your conflict of interest statement in the “Confidential to Editor” section, and submit your "Accept" recommendation.

Reviewer #1: All comments have been addressed

Reviewer #2: All comments have been addressed

2. Is the manuscript technically sound, and do the data support the conclusions?

Reviewer #1: Yes

Reviewer #2: Yes

3. Has the statistical analysis been performed appropriately and rigorously? 

Reviewer #1: Yes

Reviewer #2: Yes

4. Have the authors made all data underlying the findings in their manuscript fully available?

Reviewer #1: Yes

Reviewer #2: Yes

5. Is the manuscript presented in an intelligible fashion and written in standard English?

Reviewer #1: Yes

Reviewer #2: Yes

6. Review Comments to the Author

Reviewer #1: I thank the authors for their detailed response. All comments were adequatley addressed and I have no further comments.

Reviewer #2: The revision well addressed reviewer's comments, statistical methods are explained clearly. It is an interesting topic and potential useful method in real application, especially with open source code included in the article (link).

7. PLOS authors have the option to publish the peer review history of their article (what does this mean?). If published, this will include your full peer review and any attached files.

Reviewer #1: No

Reviewer #2: **Yes: **Changxing Ma

---

## [Editor Report · Acceptance letter]

2 Nov 2021

PONE-D-21-06687R1 

A modified weighted log-rank test for confirmatory trials with a high proportion of treatment switching 

Dear Dr. Jiménez:

I'm pleased to inform you that your manuscript has been deemed suitable for publication in PLOS ONE. Congratulations! Your manuscript is now with our production department. 

Kind regards, 

on behalf of

Dr. Farzad Taghizadeh-Hesary 

Academic Editor

PLOS ONE